# Cucumber Mosaic Virus-Induced Systemic Necrosis in *Arabidopsis thaliana*: Determinants and Role in Plant Defense

**DOI:** 10.3390/v14122790

**Published:** 2022-12-14

**Authors:** Israel Pagán, Fernando García-Arenal

**Affiliations:** Centro de Biotecnología y Genómica de Plantas UPM-INIA/CSIC and Departamento de Biotecnología-Biología Vegetal, E.T.S. Ingeniería Agronómica, Alimentaria y de Biosistemas, Universidad Politécnica de Madrid, 28045 Madrid, Spain; fernando.garciaarenal@upm.es

**Keywords:** *Arabidopsis thaliana*, cucumber mosaic virus, systemic necrosis, hypersensitive response, resistance, plant–virus coevolution

## Abstract

Effector-triggered immunity (ETI) is one of the most studied mechanisms of plant resistance to viruses. During ETI, viral proteins are recognized by specific plant R proteins, which most often trigger a hypersensitive response (HR) involving programmed cell death (PCD) and a restriction of infection in the initially infected sites. However, in some plant–virus interactions, ETI leads to a response in which PCD and virus multiplication are not restricted to the entry sites and spread throughout the plant, leading to systemic necrosis. The host and virus genetic determinants, and the consequences of this response in plant–virus coevolution, are still poorly understood. Here, we identified an allelic version of RCY1—an R protein—as the host genetic determinant of broad-spectrum systemic necrosis induced by cucumber mosaic virus (CMV) infection in the *Arabidopsis thaliana* Co-1 ecotype. Systemic necrosis reduced virus fitness by shortening the infectious period and limiting virus multiplication; thus, this phenotype could be adaptive for the plant population as a defense against CMV. However, the low frequency (less than 1%) of this phenotype in *A. thaliana* wild populations argues against this hypothesis. These results expand current knowledge on the resistance mechanisms to virus infections associated with ETI in plants.

## 1. Introduction

Plants are recurrently challenged by parasites in the wild and agroecosystems [1]. Because parasites have a negative impact on host growth and/or reproduction [2], plants have developed a variety of defense mechanisms to avoid infection and/or limit its consequences [3,4]. Plant defenses against parasites have far-reaching consequences for the fitness of both interacting organisms [5,6]. Thus, investigating their mechanistic bases is central to understanding the dynamics of plant–parasite interactions [7,8]. This is particularly true for plant viruses, which account for the largest fraction of emerging plant diseases [9].

The best-characterized plant defense against parasites, including viruses, is resistance, i.e., the host’s ability to limit parasite infection and/or multiplication [10,11]. Plant resistance involves different mechanisms: the recognition of conserved parasite-associated molecular patterns, which triggers basal defenses (PAMP-triggered immunity, PTI), and the recognition of parasite effectors that suppress PTI (effector-triggered immunity, ETI), among others [7,12,13]. During ETI, parasite effectors are recognized as avirulence (Avr) factors by specific plant R proteins. The largest class of plant *R* genes encodes a conserved nucleotide binding site (NBS), a leucine-rich repeat domain (LRR) [12,14,15] and either a coiled-coil (CC) domain or a toll-interleukin 1 receptor (TIR)-like region [16]. *R* gene-conferred resistance against plant viruses is often expressed as a hypersensitive response (HR), which results in the development of necrotic local lesions (NLLs) at infection sites that restrict virus infection [17]. In some plant–virus interactions, the expression of *R* genes results in an HR that is not limited to the infection site but expands and causes systemic necrosis [18,19,20]. This systemic necrosis is thought to be due to inefficient resistance, such that *R* genes fail to restrict the systemic spread of the virus, even if HR is induced [19]. Analyses of the effect of systemic necrosis on virus multiplication showed contrasting results. Some authors reported significant reductions [21], whereas others did not [19,20,22,23]. This diversity of results can be attributed to the differences in the moment at which virus accumulation was measured (from a few days to several weeks after infection), but the kinetics of virus multiplication in relationship with the development of systemic necrosis has not been analyzed to date. It has also been shown that restriction of NLLs in an HR reaction requires active autophagy, and impairment of autophagy through modification of autophagy-related genes results in uncontrolled necrosis. This necrosis, which is considered a pathogenic rather than defensive outcome of the initial resistance reaction, does not require virus systemic movement [24,25], adding to the complexity of the effects of systemic necrosis on virus multiplication. Thus, it is still unclear whether, and how, systemic necrosis affects virus multiplication and fitness.

From the perspective of plant–virus coevolution, systemic necrosis can be hardly viewed as selectively advantageous for the individual plant because, at odds with full resistance, fitness may drop to zero upon early infection. However, systemic necrosis could be advantageous at the host population level [20]: The effect of systemic necrosis on the virus fitness is not solely due to its effects on virus multiplication, but it may reduce transmission rates by reducing the infectious period and, for vectored viruses, the attraction and feeding behavior of insects (e.g., [26,27]). In addition, because seed dispersal can occur often at small spatial scales, genotypes responding to infection by systemic necrosis will be spatially aggregated at scales similar to those of vector dispersal [28], and infected plants will not be efficient sources of inoculum for non-resistant plants [21]. Mathematical modeling supports this hypothesis but predicts that when viruses disperse longer distances, and/or virus prevalence is high, maintaining genotypes that develop systemic necrosis under infection has little advantage for the plant population [20]. Although these authors provide compelling evidence that under certain conditions a systemic necrosis phenotype can be fixed in plant populations, it should be noted that their model assumes that plants with this “suicidal” genotype die right after infection, representing an instantaneous barrier for virus dispersal. This is not always the case, and longer times to plant death would reduce the benefits of maintaining the “suicidal” genotype in the population. Therefore, whether virus-induced systemic necrosis is evolutionarily advantageous remains under debate.

The aim of the present work is to characterize the determinants for systemic necrosis in the system formed by cucumber mosaic virus (CMV, *Bromoviridae*) and *Arabidopsis thaliana* (Brassicaceae; from now on “Arabidopsis”) and to analyze the potential consequences of systemic necrosis for virus fitness. CMV has isometric particles that separately encapsidate the three segments of a messenger-sense single-stranded RNA genome with five genes. RNA1 encodes for protein 1a. RNA2 encodes for protein 2a, which interacts with protein 1a in the viral RNA-dependent RNA replicase, and protein 2b, a suppressor of the virus-induced gene silencing resistance reaction of the plant. RNA3 encodes protein 3a, which is required for cell-to-cell movement of virus infection through the plasmodesmata, and the coat protein (CP), which is also required for cell-to-cell movement, systemic movement and aphid transmission [29]. CMV is considered a typical generalist parasite, infecting about 1200 host species in more than 100 plant families, including Arabidopsis. CMV isolates are highly diverse and are classified into two subgroups: I and II, subgroup I being further split into IA and IB, on the basis of the sequence homology of their genomes [29]. Eighteen NBS-LRR genes inducing resistance against plant viruses have been identified in Arabidopsis [17]. For example, *HRT* and *RCY1*, which are alleles of the same gene, confer HR resistance to turnip crinkle virus (TCV) and CMV, respectively [30,31]. Genetic variation of this *R* gene also allows (i) recognition of widely different parasites, as the *RPP8* gene, allelic to *HRT*/*RCY1*, triggers resistance to the oomycete *Hyaloperonospora arabidopsisdis*, and (ii) triggering different defense responses, as in the ecotype C24, point mutations in *RCY1* result in systemic necrosis in response to Y-CMV [22]. CMV is commonly found in natural populations of Arabidopsis at a prevalence of up to 80% [32], indicating that the Arabidopsis–CMV interaction is significant in nature, and that CMV infection represents a selective pressure in Arabidopsis wild populations [33]. Thus, if CMV-induced systemic necrosis is a population-level defense mechanism that reduces between-host transmission, it would be selectively advantageous for Arabidopsis and therefore relatively common in wild populations under CMV challenge. However, few ecotypes have been screened for this phenotype [34], and *RCY1*-controlled systemic necrosis has been only described in laboratory-obtained mutants rather than in wild ecotypes [20,22]. In addition, it is not known how widespread CMV isolates that activate *RCY1*-controlled systemic necrosis are and whether this plant response affects their multiplication.

Here, we report the presence of *RCY1* in the Arabidopsis Co-1 ecotype as a determinant of systemic necrosis after the infection of subgroup I CMV isolates. Systemic necrosis reduces within-host virus multiplication, affects the kinetics of virus accumulation and reduces the host infectious period. Although these results are compatible with model predictions of systemic necrosis being beneficial for the plant population, the low frequency of this genotype in Arabidopsis ecotypes argues against this hypothesis.

## 2. Materials and Methods

### 2.1. Viral Isolates and Arabidopsis Thaliana Ecotypes

The following CMV isolates were used in this work: Fny-CMV, Y-CMV, MAD99/4 and BAR96/1 belonging to subgroup IA; BAR92/1 belonging to subgroup IB; and LS-CMV belonging to subgroup II [29,35]. Fny-CMV and LS-CMV were derived from biologically active cDNA clones [36,37] by in vitro transcription with T7 RNA polymerase (New England Biolabs, Ipswich MA). Y-CMV was kindly provided by Dr. Hideki Takahashi (Tohoku University, Sendai). Isolate MAD99/4 was obtained in 1999 from an infected zucchini (*Cucurbita pepo*) plant collected in Madrid province (Spain), BAR96/1 was obtained in 1996 from an infected melon (*Cucumis melo*) plant sampled in Barcelona province (Spain) and BAR92/1 was obtained in 1992 from an infected tomato (*Solanum lycopersicum*) plant sampled in Barcelona province (Spain) [35].

For virus multiplication, transcripts of the biologically active cDNA clones of Fny-CMV and LS-CMV in 0.1 M Na_2_HPO_4_ were used to mechanically inoculate *Nicotiana clevelandii* plants. MAD99/4, BAR96/1 and BAR92/1 were multiplied from purified viral RNA previously obtained by our group [35], which was also used to infect *N. clevelandii* plants for virus multiplication. Y-CMV was obtained from infected leaf tissue, which was used to mechanically inoculate *N. clevelandii* by tissue grinding in a solution containing 0.1 M Na_2_HPO_4_, 0.5 M NaH_2_PO_4_ and 0.02% DIECA (0.01 M phosphate buffer (pH 7.0), 0.2% sodium diethyldithiocarbamate). In all cases, CMV virions were purified from infected tobacco leaves as described in [38], and viral RNA was extracted by virion disruption with phenol and sodium dodecyl sulfate [39]. All inoculations in Arabidopsis were performed with 100 ng/μL of purified viral RNA in 0.1 M Na_2_HPO_4_ when plants were at developmental stages 1.05 to 1.06 [40].

Arabidopsis plants from genotypes Co-1 (Coimbra, Portugal) and Ler (Landsberg, Poland) were used in most experiments. For plant growth, Co-1 seeds were sown on filter paper soaked with water in single plastic Petri dishes and stratified in darkness at 4 °C for 3 days before being transferred for germination to a growth chamber (22 °C, 14 h light and 70% relative humidity). Five-day-old seedlings were planted in soil-containing pots 10.5 cm in diameter and 0.43 L in volume and grown in a greenhouse (25/20 °C day/night, 16 h light).

For the screening of systemic necrosis in Arabidopsis, 100 ecotypes from the Iberian Peninsula were used (Appendix A). Ecotypes were collected from different populations and selected to cover the genetic and environmental diversity of the species in the region [41]. The Iberian Peninsula is a center of Arabidopsis genetic diversification, and the selected genotypes are representative of the species diversity in this region [42], both on the basis of the sequence homology of their genome and on the phenotypic variation upon CMV infection [33].

### 2.2. Construction of CMV Pseudorecombinants

Fny-CMV and LS-CMV biologically active cDNA clones were used to construct pseudorecombinants between these two isolates. To do so, in vitro transcripts of each genomic segment were independently obtained as described above. These transcripts were inoculated in *N. clevelandii* plants in all possible combinations (Figure 1), and then virions were purified and viral RNA was obtained as described. Plants were mechanically inoculated with purified CMV RNA (100 ng/mL) in 0.1 M Na_2_HPO_4_.

### 2.3. Time Course of CMV Multiplication in Co-1

Fny-CMV and LS-CMV were inoculated in 24 replicated Co-1 plants each. At 1, 2, 4, 6, 9, 15, 18 and 21 days post-inoculation (dpi), 0.01 g (fresh weight) of inoculated and 0.01 g (fresh weight) of systemically infected rosette leaves were harvested, with three replicates per time point. In each sample, CMV multiplication was quantified as viral RNA accumulation as described in [43]. The time interval was selected to cover the variability in the development of systemic necrosis (at 21 dpi all Fny-CMV-infected Co-1 plants had developed this phenotype, see Section 3).

### 2.4. Use of Molecular Markers to Detect RCY1 in Arabidopsis Genomic DNA

To identify the presence of the *RCY1* gene, PCR amplifications with DNA Taq polymerase (Biotools, Madrid, Spain) were performed using the plant DNA purified from 0.01 g of leaf material homogenized in EB buffer (2% CTAB, 1.4M NaCl, 20 mM EDTA and 100 mM Tris) followed by partition in chloroform [44]. Primers RCY1-F (5′-CAAAGTCCAACACATTCCCGA-3′) and RCY1-R (5′-CACAACATAACGATGCACTGAAAGC-3′), designed using the *RCY1* nucleotide sequence of Arabidopsis ecotype C24 (Acc. No. AB087829) and the *RPP8* sequence of Arabidopsis ecotype Ler (Acc. No. AF089710), were used. PCR products were separated by electrophoresis in 2.5% agarose in TAE [45].

The presence of nga129 and CIW9 microsatellites, located at 6 cM upstream and downstream of the *RCY1* gene, respectively, were also analyzed. Both microsatellites are polymorphic in Arabidopsis and allow differentiating between Ler and Co-1. The nga129 microsatellite was amplified by PCR using primers nga129-F (5′-TCAGGAGGAACTAAAGTGAGGG-3′) and nga129-R (5′-CACACTGAAGATGGTCTTGAGG-3′). The CIW9 microsatellite was amplified using primers CIW9-F (5′-CAGACGTATCAAATGACAAATG-3′) and CIW9-R (5′-GACTACTGCTCAAACTATTCGG-3′). PCR products were separated as above.

### 2.5. Sequencing of the RCY1 Gene in Co-1 Plants

To obtain the complete sequence of the *RCY1* gene of the Co-1 ecotype, five primer pairs based on the *RCY1* nucleotide sequence from C24 were designed in such a way that adjacent fragments overlapped by at least 150 nt (Appendix A). PCR products were separated by electrophoresis in 1% agarose in TAE. Amplicons of the expected size were gel purified using a QIAquick Gel Extraction kit (Qiagen, Valencia, CA, USA) and sequenced in an ABIprism A310 (Applied Biosystems, Foster City, EEUU) sequencer using the corresponding primer pairs. Chromatograms were read using Chromas 2.5 (Technelysium, South Brisbane, Australia). To obtain the complete nucleotide sequence of the *RCY1* gene, fragments were aligned using Muscle 3.8 [46]. Nucleotide identity in overlapping regions of adjacent fragments was found to be 100%. The same software was used to align the *RCY1* nucleotide sequence of Co-1 and C24. Amino acid RCY1 sequences of these two ecotypes were also aligned using Muscle 3.8. Average nucleotide and amino acid identity values between the overlapping fragments of Co-1 *RCY1* PCR products were estimated as the nucleotide–amino acid substitution/total gene length [47] using pANIto [https://github.com/sanger-pathogens/panito (Accessed on 14 November 2022)]. The sequence of the *RCY1* version of Co-1 has been deposited in GenBank under Acc. No. OP991902.

### 2.6. Statistical Analyses

Virus multiplication and the kinetics of systemic necrosis were normally distributed, and variances were homogeneous according to Kolmogorov–Smirnov and Levene’s tests, respectively. Therefore, they were fitted to a normal distribution, and differences between plant ecotypes were analyzed by general linear models (GLMs) considering the Arabidopsis ecotype or the virus isolate as fixed factors. The segregation of the CMV-induced systemic necrosis in F2 plants derived from the Co-1 x Ler crossing was analyzed using Fisher exact tests [48]. Statistical analyses were conducted using R v.3.6.3 [49].

## 3. Results

### 3.1. Genetic Determinants in CMV of Systemic Necrosis in Arabidopsis Co-1

We have reported that systemic necrosis was induced in Arabidopsis Co-1 ecotype after infection by CMV isolates Fny-CMV and De72-CMV in subgroup IA, but not by isolate LS-CMV in subgroup II [43]. Thus, pseudorecombinants between Fny-CMV and LS-CMV were generated to map the determinants of this host reaction (Figure 1). Viral RNA of each CMV pseudorecombinant was used to inoculate Co-1 plants with ten replicates per pseudorecombinant. Another ten Co-1 plants were inoculated with Fny-CMV (positive control) and LS-CMV (negative control). In plants infected by Fny-CMV or by pseudorecombinants F1L2F3, L1L2F3 and L1F2F3, NLLs developed in inoculated leaves 3 days post-inoculation (dpi) that did not remain localized but expanded until the complete necrosis of the inoculated leaves at 7 dpi. No NLL or other necrosis was observed in leaves inoculated with LS-CMV and with pseudorecombinants F1F2L3, F1L2L3 and L1F2L3 (Figure 2).

Plants infected by Fny-CMV or by pseudorecombinants F1L2F3, L1L2F3 and L1F2F3 started to develop systemic necrosis two weeks after inoculation, which led to plant death a week later (Figure 3). In contrast, plants infected by LS-CMV, F1F2L3, F1L2L3 and L1F2L3 developed symptoms of leaf curl and lamina reduction and stunting of the inflorescence (Figure 3). The experiment was repeated three times with the same result. Thus, the genetic determinant of CMV systemic necrosis in Arabidopsis Co-1 plants is located in RNA3.

### 3.2. Systemic Necrosis Reduces CMV Multiplication

The level of CMV accumulation in inoculated and systemically infected rosette leaves in Co-1 plants between 1 and 21 dpi was monitored; thus, the time span in which all infected plants developed the systemic necrosis was covered. In inoculated leaves, Fny- and LS-CMV accumulation was similar up to 2 dpi (*F*_1,17_ = 0.03, *p* = 0.857) and was much higher for the latter isolate afterward (*F*_1,11_ = 31.73, *p* < 1 × 10^−4^) (Figure 4). The maximum of Fny-CMV accumulation was reached at 2 dpi, whereas for LS-CMV it was detected at 4 to 6 dpi (Figure 4A). Similarly, in systemically infected leaves, the multiplication of both isolates was similar up to 6 dpi (*F*_1,29_ = 1.03, *p* = 0.320), and the multiplication was much higher for LS-CMV than for Fny-CMV from that point to the end of the monitored period (*F*_1,24_ = 45.80, *p* < 1 × 10^−4^). Interestingly, the peak of Fny-CMV multiplication occurred at 18 dpi, whereas that of LS-CMV was observed at 9 dpi (Figure 4B). The observed difference in virus multiplication might be due to isolate-specific effects rather than host ecotype effects associated with systemic necrosis. Indeed, on average, LS-CMV shows higher multiplication than Fny-CMV across Arabidopsis ecotypes [43]. Thus, differences observed in our time course experiment were compared with average differences between 18 Arabidopsis ecotypes derived from data reported in [43]. Both experiments were performed under the same conditions, plants were inoculated at the same phenological stage, and data obtained at the same time post-inoculation were compared. Results indicated that, while LS-CMV multiplication in Arabidopsis at 15 dpi was on average 3-fold higher than that of Fny-CMV, this difference increased to 9-fold in Co-1 (*F*_1,202_ = 4.35, *p* = 0.038). Moreover, Co-1 was included in the mentioned set of 18 ecotypes, and the reported virus accumulation and ours are in the same range (7.2 times higher for LS-CMV in [43] and 9.1 times higher here (Figure 4)). These observations strongly suggest that the systemic necrosis reduces CMV multiplication.

### 3.3. Systemic Necrosis in Co-1 Is Induced by Subgroup I CMV Isolates

Takahashi et al. [34] described that the Arabidopsis ecotype C24 was resistant to the yellow strain of cucumber mosaic virus (Y-CMV, subgroup IA) through the activation of an HR. This response was exclusive to Y-CMV and was not triggered by other isolates of subgroup IA. The phenotype observed in Co-1 plants differed from the one described by [34] in that the initial necrotic local lesions lead to systemic necrosis, and in that it is triggered by more than one isolate of subgroup IA: Fny-CMV and De72-CMV in our previous work [43].

To test if this differential response was due to differences in virus determinants, ten Co-1 plants were inoculated with two additional CMV isolates belonging to subgroup IA (MAD99/4, BAR96/1) and with Y-CMV, and with one subgroup IB isolate, BAR92/1 (Figure 5). Co-1 plants were infected by all the CMV isolates tested. In all cases, NLLs were observed in inoculated leaves at 3 dpi, and systemic necrosis was observed at 15 dpi. This experiment was repeated twice with the same results. Thus, the virus-induced systemic necrosis in Co-1 is a general response to subgroup I isolates, including Y-CMV.

Takahashi et al. [50] identified the CMV genetic determinant of this HR response to be the CP encoded in RNA3. Following this work, Abebe et al. [20] have reported that mutation T45M in the Y-CMV coat protein determines a systemic infection and necrosis in C24 plants, rather than the HR induced by wild-type Y-CMV. We checked if mutation T45M could have arisen in Y-CMV CP during propagation in *N. clevelandii* plants, and if an M at position 45 was present in the CP of subgroup I isolates inducing systemic necrosis in this work. For this, the nucleotide sequence of the CP gene of all these isolates, obtained by [35], was retrieved from GenBank. Sequence alignment indicated that the CP of our Y-CMV was identical to the original one provided by Dr. Takahashi (Acc. No. M57602) and that none of the other four subgroup I isolates had an M at position 45 (Appendix A).

### 3.4. Genetic Determinants in Arabidopsis of CMV-Induced Systemic Necrosis

As a first step in the analysis of Co-1 determinants of the systemic necrosis reaction, the inheritance of this trait was determined. To do so, we inoculated Fny-CMV in 174 plants of the F2 derived from a Co-1 x Ler crossing. All inoculated plants were infected, and systemic necrosis developed in 128 (73.56%), while the infection induced leaf curl and lamina reduction in 46 (26.44%). Thus, the systemic necrosis phenotype had a 3:1 segregation (*χ*^2^ = 0.14, *p* = 0.712), indicating its control by a single dominant gene.

In the same F2 plants, the kinetics of systemic necrosis was also analyzed by quantifying two traits: the percentage of plants showing systemic necrosis at different times post-inoculation, as indicative of the onset of necrosis, and the number of days from the appearance of systemic necrosis to plant death, as indicative of the speed of necrosis development. Both traits were also quantified in ten Co-1 parental plants as a reference. The percentage of plants showing systemic necrosis increased until 21 dpi, when 75% of the F2 plants and 100% of the Co-1 plants developed this symptom (Figure 6A). The onset of systemic necrosis was faster in the F2 plants than in the Co-1 parentals (7 vs. 14 dpi), with 25% of F2 plants showing evidence of this symptom before any of the Co-1 plants did (Figure 6A). Thus, the onset of systemic necrosis appears to be a quantitative trait showing transgressive segregation. The speed of necrosis development was also a quantitative trait (Figure 6B). GLM analyses indicated that there was not a significant difference in the speed at which the systemic necrosis developed between F2 and Co-1 parental plants (*F*_1,139_ = 0.03, *p* = 0.856), with an average of 5.66 and 5.60 days for F2 and Co-1 plants, respectively. However, it should be noted that the span of days to complete systemic necrosis was narrower for Co-1 plants (3 to 8 days) than for F2 plants (2–10 days) (Figure 6B).

Takahashi et al. [31] described that the activation of the HR to Y-CMV in Arabidopsis ecotype C24 was controlled by *RCY1*. Given that Fny-CMV infection induced the formation of NLLs well before the beginning of the systemic necrosis, that this trait was determined by a single dominant gene and that recognition involved the same genomic segment of CMV [50], the potential role of RCY1 in the development of systemic necrosis was explored.

PCR detection of the Co-1 *RCY1* allele and of the Ler *RPP8* allele was performed with primers that amplified a 343 nt (nucleotides 5044 to 5387) fragment for *RCY1* and a fragment of 382 nt (from nucleotide 4151 to nucleotide 4533) for *RPP8*, allowing the identification of the two alleles both in homozygosity and in heterozygosity (Appendix A). PCR analyses were performed for the 174 plants of the Co-1 x Ler cross. Out of the 128 F2 plants showing systemic necrosis upon Fny-CMV infection, 33 had the *RCY1* allele in homozygosis and 95 had the *RCY1* allele in heterozygosis, and of the 46 plants that did not develop systemic necrosis, 44 had the *RPP8* allele in homozygosis and 2 had it in heterozygosis. Thus, co-segregation indicated that the *RCY1* gene is required for the systemic necrosis phenotype.

However, the fact that two individuals had the *RCY1* allele but did not develop systemic necrosis suggested that a second gene could be involved in the control of this phenotype. To address this possibility, the association of two microsatellites (nga129 and CIW9), which flank *RCY1* at 6 cM at each side, with the development of systemic necrosis upon Fny-CMV infection was analyzed. The primers used for nga129 yielded amplicons of 179 nucleotides in Ler and of 170 nucleotides in Co-1, and those for CIW9 yielded amplicons of 145 nt in Ler and of 170 nt in Co-1. The difference in the amplicon size allowed identifying the two alleles both in homozygosity and in heterozygosity (Appendix A). Both microsatellites showed a similar segregation (*χ*^2^ = 1.36, *p* = 0.929), indicating that they are at the same distance from the gene that controls the systemic necrosis (Table 1). This result suggests that *RCY1* is the only gene involved in the development of the symptom and eliminates the possibility that other host genes regulate the response triggered by RCY1.

Since *RCY1* was identified as the determinant of Co-1 systemic necrosis upon infection by CMV subgroup I isolates, while *RCY1* determines an HR response exclusively upon infection by Y-CMV in ecotype C24, the possibility that Co-1 and C24 had different alleles of *RCY1* was explored. To do so, the complete sequence of the Co-1 *RCY1* was obtained and compared with that of C24 (Figure 7). Thirty-six non-synonymous nucleotide substitutions (i.e., resulting in amino acid changes) occurred in the Co-1 *RCY1* coding region as compared with that of C24 *RCY1*. All these changes were located within the coding region approximately spanning nucleotides 650 to 906 and therefore within the leucine-rich repeat domain (LRR) (Figure 7).

### 3.5. Screening of the Frequency of CMV-Induced Systemic Necrosis in the Arabidopsis Population of the Iberian Peninsula

To analyze the frequency of determinants of systemic necrosis in reaction to CMV infection, seven individuals of 100 Arabidopsis ecotypes from the Iberian Peninsula were inoculated with Fny-CMV. All plants developed typical symptoms of virus infection, including different degrees of leaf curl and lamina reduction in the rosettes (Figure 8) and stunting of the reproductive structures. None of the tested ecotypes developed NLLs and/or systemic necrosis upon Fny-CMV infection, indicating that determinants of these phenotypes are rare in the Arabidopsis population.

## 4. Discussion

Resistance, together with tolerance [51], is one of the most relevant, effective and widespread defenses of plants against viruses. Thus, during the last few decades, understanding the molecular bases of plant resistance to viruses has been a long-standing goal in plant pathology [3,4]. ETI, resulting from the recognition of viral Avr proteins by plant R proteins, restricts virus infection to the entry sites in most cases, preventing systemic spread through the activation of an HR [3,17]. In contrast with the extensive understanding of how viruses induce HR, much less is known about other phenotypes resulting from Avr-R recognition. One such phenotype is virus-induced plant systemic necrosis, which does not localize virus infection and results in the death of the infected plant. To contribute to a better understanding of the role of systemic necrosis in plant–virus interactions, we characterize here the host and virus determinant of this phenotype in the Arabidopsis Co-1-CMV interaction, and we explore its potential role in the evolution of this pathosystem.

Our work shows that systemic necrosis in Co-1 is triggered by the recognition of a protein encoded in RNA3 of subgroup I of CMV isolates, either the movement protein or the CP. Recognition results in an HR response that fails to localize the necrosis to the initial infection foci but propagates systemically until the infected plant dies. Interestingly, the onset of systemic necrosis, at 14 dpi, is much delayed compared to the appearance of NLLs in inoculated leaves at 6 dpi, which can be explained if (i) the virus systemic spread is delayed during necrosis at inoculated leaves, later triggering necrosis in newly invaded organs, or (ii) the systemic necrosis is activated by a long-distance signal that is produced in inoculated leaves. In this regard, failure in the restriction of NLL resulting in virus-induced systemic necrosis has been associated with changes in the levels of salicylic acid (SA) [52], which acts as a mobile signal [53], and with abnormal function of autophagy-related proteins [54], which induces necrosis in uninfected cells [24]. The analysis of the molecular mechanisms that control systemic necrosis downstream of HR is out of the scope of this work and represents an interesting avenue for future research. However, in support of a link between systemic necrosis and virus colonization, rather than a systemic signal, we found that systemic necrosis and CMV multiplication in systemically infected leaves are simultaneous (compare Figure 4B and Figure 6A), whereas the maximum accumulation of CMV isolates that do not induce NLL or systemic necrosis occurs much earlier.

We identified *RCY1* and RNA3 as the host and virus determinants of systemic necrosis in Co-1. These are also the determinants of HR in ecotype C24 upon Y-CMV infection [31,34,50]. Despite the plant and virus genetic determinants being the same in the Y-CMV/C24 and in the subgroup I CMV/Co-1 interactions, the observed phenotypes and the specificity of recognition by R are different. Our results show that these differences in phenotype/specificity may be explained by differences in the *RCY1* sequence of Co1 and C-24, particularly in the LRR domain, rather than by differences in previously described determinants of systemic necrosis between the CMV isolates considered in our work. This conclusion is consistent with point mutations in the LRR domain of C24 RCY1 determining a systemic necrosis phenotype in response to Y-CMV infection [22]. Interestingly, the LLR domain is thought to be involved in the degradation of RCY1 right after triggering HR, a process that allows controlling the size of NLLs [52,55]. We may speculate that deletions in the LRR domain of Co-1 RCY1 relative to C24 RCY1 interfere with RCY1 degradation and restriction of necrosis to NLLs.

Regardless of the mechanisms involved, our results suggest that systemic necrosis is a resistance reaction, as it reduces virus accumulation. In addition to the results presented in Figure 4, our previous work had shown that differences in accumulation between LS-CMV, which does not induce systemic necrosis in Co-1, and Fny-CMV, which does, are higher in Co-1 than in most Arabidopsis ecotypes [43]. An effect of systemic necrosis on virus accumulation is apparently at odds with results reported by the authors [22], who did not find a significant difference in CMV multiplication in non-inoculated leaves between plants developing or not developing *RCY1*-controlled systemic necrosis, and the authors of [20], who reported a small decrease in the multiplicity of infection of CMV isolates inducing systemic necrosis in inoculated leaves as compared to those that did not. Discrepancies with these two reports may be due to the different times post-inoculation in which virus accumulation was quantified: up to 7 days in systemically infected leaves and 1 day post-inoculation in inoculated ones. At such early times post-infection, our data also show little variation in the level of LS- and Fny-CMV accumulation (see Figure 4). A reduction in virus multiplication associated with systemic necrosis has also been reported in Arabidopsis plants with the *TuNI* gene that determines systemic necrosis in response to infection by turnip mosaic virus (TuMV), where virus accumulation was quantified two weeks after inoculation [21].

As a resistance reaction, systemic necrosis reduces subgroup I CMV titer in systemically infected leaves, which will translate into a reduction in virus transmission: it is well established that the level of virus multiplication correlates positively both with aphid [26,56] and with seed transmission rates [57]. Horizontal transmission will be further decreased by systemic necrosis because of a reduction in the plant lifespan and thus in the infectious period. Hence, our results indicate that systemic necrosis has a direct effect on the virus fitness, suggesting that this phenotype might be the result of the adaptive evolution of the plant in response to CMV infection pressure. The effects of systemic necrosis of Co-1 to CMV are compatible with the hypothesis that systemic necrosis is a defense that acts at the population, rather than at the individual, level through a sort of plant “suicide” strategy that would reduce the probability of virus transmission to adjacent plants [20]. Given that in many plant species, including Arabidopsis, genetically related individuals grow in proximity due to short-distance seed dispersal, this suicidal mechanism would increase plant fitness and therefore be adaptive [20,21]. Although poorly studied, systemic necrosis might not be a rare plant response to virus infection as, in addition to Arabidopsis and CMV, it has been reported in the TuMV–Arabidopsis, *Potato virus Y*–tobacco, *Soybean mosaic virus*–soybean and *Plantago asiatica mosaic virus–N. benthamiana* interactions [21,23,58,59]. Using mathematical modeling, Abebe et al. [20] showed that maintenance of the systemic necrosis phenotype in plant populations requires the following: (i) virus dispersal must primarily occur at short distances (a few meters), and (ii) virus prevalence in the plant population must be low. These conditions are not met in the Arabidopsis–CMV system: the genetic diversity of CMV is structured at the regional rather than the local scale [60], which argues against transmission being mainly at short distances, and CMV incidence in wild Arabidopsis populations is high [32]. Accordingly, we show that the frequency of the systemic necrosis phenotype is extremely low in Arabidopsis wild populations, which is not compatible with a significant role in resistance to CMV. Indeed, our work has shown that the defense of Arabidopsis wild populations against CMV depends primarily on quantitative rather than qualitative defenses, both resistance and tolerance, which are regarded as the most common defenses of plants against viruses [33]. A detailed experimental study of the consequences of systemic necrosis for virus epidemiology will contribute to clarifying the role of this host response in plant–virus coevolution.

In summary, the present study, based on the characterization of the systemic necrosis phenotype of a wild ecotype of Arabidopsis in response to an ample spectrum of CMV isolates, contributes to understanding the underlying mechanisms and shows that systemic necrosis is a resistance reaction. However, the benefits of systemic necrosis for Arabidopsis fitness under CMV infection do not seem to suffice for this phenotype being frequent in wild populations of this host, which calls into question its adaptive value in this plant–virus system.

## Figures and Tables

**Figure 1 viruses-14-02790-f001:**
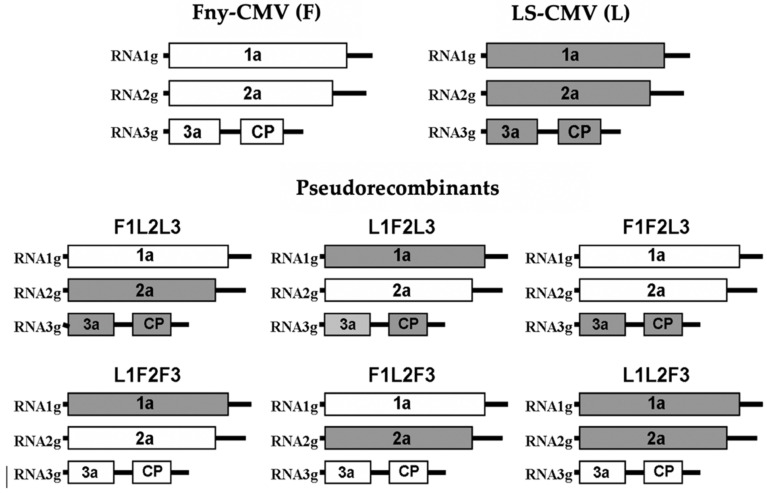
Pseudorecombinants between Fny-CMV and LS-CMV used to map the genetic determinants of systemic necrosis in Arabidopsis Co-1 plants.

**Figure 2 viruses-14-02790-f002:**
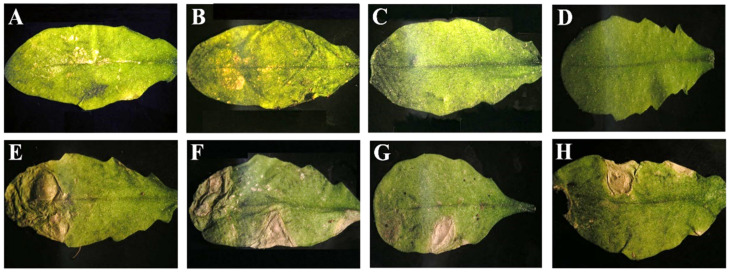
Co-1 leaves inoculated with each of the six pseudorecombinants between Fny-CMV and LS-CMV and with the two wild-type isolates. (**A**–**D**) Leaves inoculated with LS-CMV (**A**), F1L2L3 (**B**), L1F2L3 (**C**) and F1F2L3 (**D**). (**E**–**H**) NLLs in leaves inoculated with Fny-CMV (**E**), L1F2F3 (**F**), F1L2F3 (**G**) and L1L2F3 (**H**).

**Figure 3 viruses-14-02790-f003:**
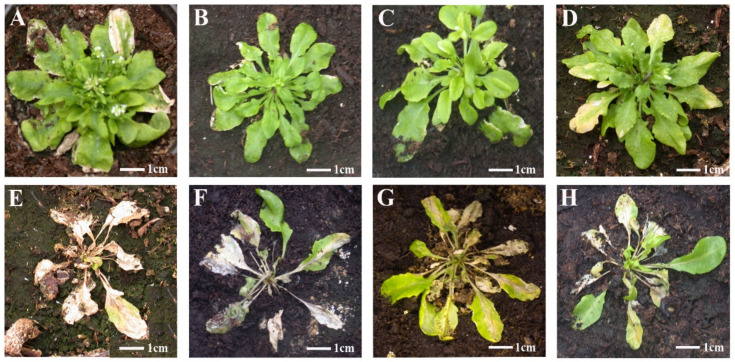
Symptoms in Co-1 plants inoculated with each of the six pseudorecombinants between Fny-CMV and LS-CMV and with the two wild-type isolates. (**A**–**D**) Leaf curl and lamina reduction in plants infected by LS-CMV (**A**), F1L2L3 (**B**), L1F2L3 (**C**) and F1F2L3 (**D**). (**E**–**H**) Systemic necrotic in plants infected by Fny-CMV (**E**), L1F2F3 (**F**), F1L2F3 (**G**) and L1L2F3 (**H**).

**Figure 4 viruses-14-02790-f004:**
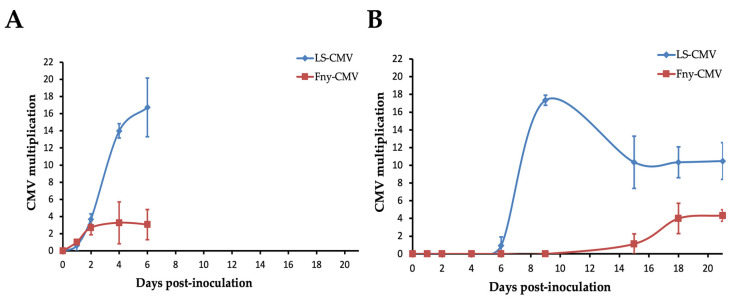
Time course of CMV multiplication in Co-1. CMV multiplication is estimated from the accumulation of viral RNA (mg/g fresh weight) of (**A**) inoculated rosette leaves and (**B**) systemically infected rosette leaves. Data for LS-CMV are represented as blue diamonds and data for Fny-CMV are represented as red squares. Data are mean ± standard error of three replicates per time point and virus.

**Figure 5 viruses-14-02790-f005:**
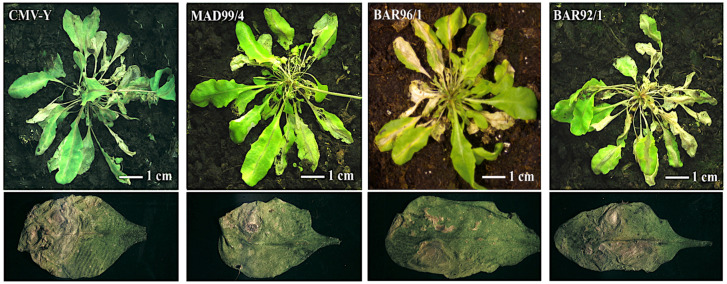
Symptoms of virus infection in Co-1 plants inoculated with different CMV isolates. MAD99/4, BAR96/1 and Y-CMV belong to subgroup IA, and BAR92/1 belongs to subgroup IB.

**Figure 6 viruses-14-02790-f006:**
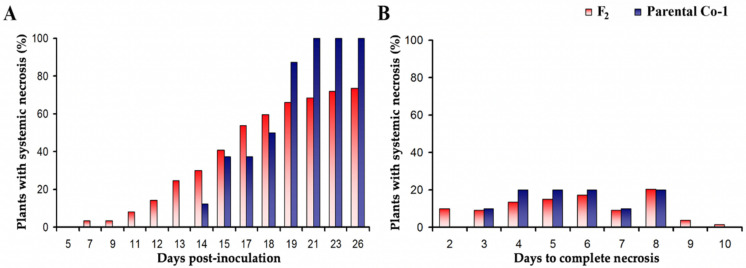
Speed in the onset and in the development of systemic necrosis in Arabidopsis. (**A**) Percentage of F2 and Co-1 parental plants showing systemic necrosis according to days after Fny-CMV inoculation. (**B**) Percentage of F2 and Co-1 parental plants according to the number of days from onset of systemic necrosis to plant death (complete necrosis).

**Figure 7 viruses-14-02790-f007:**
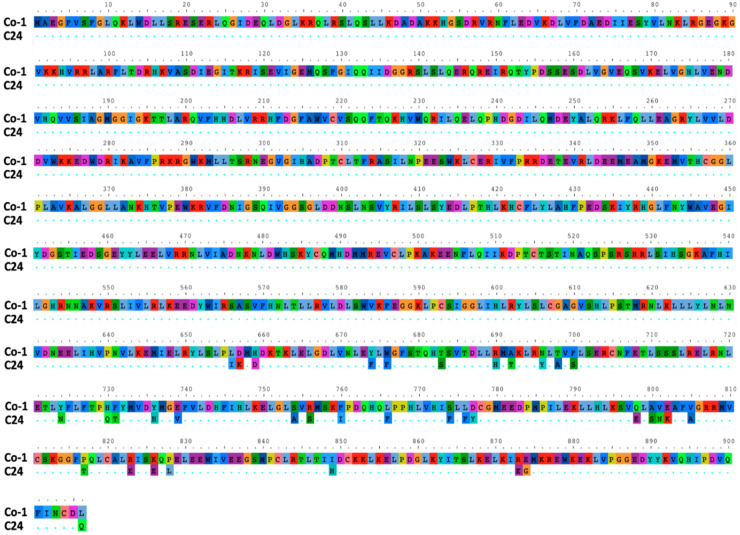
RCY1 amino acid sequence in which differences between Co-1 and C24 are highlighted. Dots in the C24 sequence indicate the same amino acid in both RCY1 proteins.

**Figure 8 viruses-14-02790-f008:**
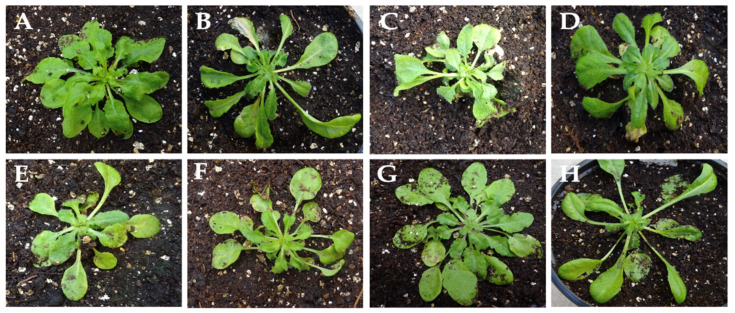
Symptoms of Fny-CMV in Arabidopsis ecotypes of the Iberian Peninsula: (**A**) Bis-0; (**B**) Cal-0; (**C**) Cor-0; (**D**) Fun-0; (**E**) Gra-0; (**F**) Gud-3; (**G**) Tor-1; (**H**) Ven-0 (see Appendix A).

**Table 1 viruses-14-02790-t001:** Segregation of the nga129 and CIW9 microsatellites in the Ler x Co-1 F2.

Microsatellite	Symptom	*RPP8* Allele	Heterozygous	*RCY1* Allele
nga 129				
	Systemic necrosis	8	61	22
	Lamina reduction	14	5	1
CIW 9				
	Systemic necrosis	10	62	20
	Lamina reduction	13	5	0

## Data Availability

Data available as Appendix A.

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
