# Peer review of "Cucumber Mosaic Virus-Induced Systemic Necrosis in Arabidopsis thaliana: Determinants and Role in Plant Defense"

_viruses, 2022, doi:10.3390/v14122790_

Round 1

Reviewer 1 Report

This MS describes a study on a systemic necrosis phenotype of Arabidopsis after CMV infection. The genetic determinant of the systemic necrosis was mapped to the RNA3 of CMV and the RCY1 of Arabidopsis. The systemic necrosis restricts the multiplication of CMV and by shortening the lifespan of the host, may also restrict the transmission of CMV. It seems that this phenotype is evolutionarily advantageous to Arabidopsis at the population level. However, the authors find that this systemic necrosis phenotype is rare in natural populations of Arabidopsis. This is interesting, although not surprising given the prevalence of CMV in nature. Overall, the MS is well written. I have only a minor suggestion: the authors may consider making the MS more concise by trying to tell a more straightforward story.

Others

Line 304 derive into

Line 427 included

Author Response

Thank you for your comments, and those of two reviewers, on our manuscript ‘Cucumber mosaic virus-induced systemic necrosis in Arabidopsis thaliana: determinants and role in plant defense’ (ID: viruses-2100348). Following your advice, we are resubmitting a new version of the paper in which the reviewers’ comments have been fully considered. Below, we address reviewers´ comments individually.

Reviewer 1

This MS describes a study on a systemic necrosis phenotype of Arabidopsis after CMV infection. The genetic determinant of the systemic necrosis was mapped to the RNA3 of CMV and the RCY1 of Arabidopsis. The systemic necrosis restricts the multiplication of CMV and by shortening the lifespan of the host, may also restrict the transmission of CMV. It seems that this phenotype is evolutionarily advantageous to Arabidopsis at the population level. However, the authors find that this systemic necrosis phenotype is rare in natural populations of Arabidopsis. This is interesting, although not surprising given the prevalence of CMV in nature. Overall, the MS is well written. I have only a minor suggestion: the authors may consider making the MS more concise by trying to tell a more straightforward story.

Response: We thank the reviewer for her/his comments on our work. Following her/his advice, we made several changes in the Introduction and Discussion sections to try to make the manuscript more precise. For instance, we have clearly stated the aim of the study and we have removed some parts of the Discussion that were redundant with the Introduction. We have also modified Figure 7 to present a more clearer view of the amino acid differences between the RCY1 versions of C24 and Co-1.

Others

Line 304 derive into

Line 427 included

Response: Both corrected.

Reviewer 2 Report

Dear Authors,

I have an opportunity to review manuscript entitled “Cucumber mosaic virus-induced systemic necrosis in Arabidopsis thaliana: determinants and role in plant defense” submitted to Viruses MDPI Journal.

Authors concentrated on identification of  an allelic version of RCY1 – an R protein – as the host genetic determinant of broad-spectrum systemic necrosis induced by cucumber mosaic virus (CMV) infection in the Arabidopsis thaliana Co-1 ecotype.

Moreover, Authors stated that systemic necrosis reduced virus fitness by shortening the infectious period and limiting virus multiplication and, thus, this phenotype could be adaptive for the plant population as a defense to CMV despite of the low frequency (less than 1%) of this phenotype in A. thaliana wild populations.

Article is basically well written and add knew findings in understanding of plant response to CMV and other Bromoviridae;

Introduction give the reader sufficient background to analyze obtained results, but Please, underline clearly the aim of the study;

The material and methods are clearly described in a repetitive way;

The results are clearly presented and generates in general no questions;

Figure 6 – please, explain what does it mean days to complete necrosis ?

I have a question, whether the authors checked also any physiological parameters typical of HR?

The end-part of discussion presnted well-organised summarize and I have a question: Can Authors add some future prospects coming from obtained results?;

Sincerely

Author Response

Dear editor,

Thank you for your comments, and those of two reviewers, on our manuscript ‘Cucumber mosaic virus-induced systemic necrosis in Arabidopsis thaliana: determinants and role in plant defense’ (ID: viruses-2100348). Following your advice, we are resubmitting a new version of the paper in which the reviewers’ comments have been fully considered. Below, we address reviewers´ comments individually.

Reviewer 2

I have an opportunity to review manuscript entitled “Cucumber mosaic virus-induced systemic necrosis in Arabidopsis thaliana: determinants and role in plant defense” submitted to Viruses MDPI Journal. Authors concentrated on identification of an allelic version of RCY1 – an R protein – as the host genetic determinant of broad-spectrum systemic necrosis induced by cucumber mosaic virus (CMV) infection in the Arabidopsis thaliana Co-1 ecotype. Moreover, Authors stated that systemic necrosis reduced virus fitness by shortening the infectious period and limiting virus multiplication and, thus, this phenotype could be adaptive for the plant population as a defense to CMV despite of the low frequency (less than 1%) of this phenotype in A. thaliana wild populations. Article is basically well written and add new findings in understanding of plant response to CMV and other Bromoviridae. The material and methods are clearly described in a repetitive way. The results are clearly presented and generates in general no questions.

Response: We thank the reviewer for the positive view of our manuscript, and we are glad that the reviewer finds our work of interest.

Introduction give the reader sufficient background to analyze obtained results, but Please, underline clearly the aim of the study;

Response: We added a sentence stating the main aim of the work (Lines 86-89).

Figure 6 – please, explain what does it mean days to complete necrosis?

Response: We clarified this point in the legend of Figure 6.

I have a question, whether the authors checked also any physiological parameters typical of HR?

Response: The reviewer makes an interesting question here. Testing Co-1 physiological changes under CMV infection was out of the scope of this work (as acknowledged in lines 521-523 of the original version), but it is an interesting avenue of research that we may explore in the near future. Accordingly, we have added a comment on this regard as part of the answer to the next reviewer´s comment.

The end-part of discussion presented well-organised summarize and I have a question: Can Authors add some future prospects coming from obtained results?

Response: We thank the reviewer for her/his suggestion. We have commented the most interesting aspects to be addressed in future research (Lines 526 and 610-612).